# PREDICTOR-CORRECTOR ALGORITHMS FOR STOCHASTIC OPTIMIZATION UNDER GRADUAL DISTRIBUTION SHIFT

**Subha Maity**
Department of Statistics,
University of Michigan,
smaity@umich.edu

**Debarghya Mukherjee**
Department of Operations Research
& Financial Engineering,
Princeton University,
dm8341@princeton.edu

**Moulinath Banerjee**
Department of Statistics,
University of Michigan,
moulib@umich.edu

**Yuekai Sun**
Department of Statistics,
University of Michigan,
yuekai@umich.edu

## ABSTRACT

Time-varying stochastic optimization problems frequently arise in machine learning practice (*e.g.* gradual domain shift, object tracking, strategic classification). Often, the underlying process that drives the distribution shift is continuous in nature. We exploit this underlying continuity by developing predictor-corrector algorithms for time-varying stochastic optimization that *anticipates* changes in the underlying data generating process through a predictor-corrector term in the update rule. The key challenge is the estimation of the predictor-corrector term; a naive approach based on sample-average approximation may lead to non-convergence. We develop a general moving-average based method to estimate the predictor-corrector term and provide error bounds for the iterates, both in presence of pure and noisy access to the queries from the relevant derivatives of the loss function. Furthermore, we show (theoretically and empirically in several examples) that our method outperforms non-predictor corrector methods that do not anticipate changes in the data generating process. [1]

## 1 INTRODUCTION

Stochastic optimization is a basic problem in modern machine learning (ML) theory and practice. Although there is a voluminous literature on stochastic optimization (Agarwal et al., 2014; Moulines & Bach, 2011; Bottou, 2003; 2012; Bottou & Bousquet, 2007), most prior works consider a *time-invariant* stochastic optimization problem in which the data generating distribution is not changing over time. However, there is an abundance of real examples in which the underlying optimization problem is time-varying which can be broadly divided into two categories: the first kind arises due to exogenous variation in the data generating process. A concrete example is the object tracking problem in which an observer observes (noisy) signals regarding the position of a moving object, and the goal is inferring the trajectory of the object. The second kind of time-varying optimization problem arises due to endogenous variation in the data generating process. Examples here include strategic classification (Dong et al., 2018; Hardt et al., 2016) and performative prediction (Perdomo et al., 2020; Mendler-Dünner et al., 2020; Brown et al., 2022). Although there are a few recent papers on time-varying stochastic optimization (e.g., Cutler et al. (2021); Nonhoff & Müller (2020); Dixit et al. (2019; 2018)), they model the temporal drift as discrete, precluding them from exploiting the smoothness in the drift. This leads to worse asymptotic tracking error, depending on the magnitude of the temporal drift of the optimal solution, (*e.g.* see Popkov (2005). Zavlanos et al. (2012), Zhang et al. (2009), Ling & Ribeiro (2013) and references therein).

---

[1]Codes: https://github.com/smaityumich/concept-drift.

In this paper, we focus on time-varying stochastic optimization problems in which the temporal drift is driven by a *continuous-time* process. We leverage the smoothness of the temporal drift to develop predictor-corrector (PC) methods for stochastic optimization that *anticipates* future changes in the data generating process to improve the asymptotic tracking error (ATE) (see Definition 2.1). The main benefit of such methods is smaller tracking error (compared to other stochastic optimization algorithms that does not leverage smoothness of the temporal drift). A primary challenge in stochastic time-varying optimization is properly accounting for the temporal drift through a PC term in the update rule. The noise in the stochastic setting makes naive estimates of the PC term unstable, and may lead to non-convergence. One of our main contributions is developing a general way of estimating the PC term.

We complement the methodological contributions with theoretical results that show PC stochastic optimization algorithms inherit the benefits of their non-stochastic counterparts for time-varying problems. In particular, we show that PC stochastic optimization algorithms have smaller asymptotic tracking error (ATE) compared to their non-PC counterparts. We also demonstrate the superiority of PC algorithms empirically in a time-varying linear regression and target tracking applications. The rest of the paper is organized as follows: In Sections 2 and 3, we present the algorithm and highlight the difference between predictor-corrector based algorithm and time non-adaptive algorithms like simple gradient descent. We further present theories regarding the bound on ATE of the algorithms of both kinds. In Section 4, we present three concrete instances of time-varying stochastic optimization problems driven by underlying gradual distributions shifts and derive the details of PC algorithms for these problems. Section 5 concludes.

## 2 FRAMEWORK AND ALGORITHM

In a typical learning problem, we have $n$ samples $X_1, \ldots, X_n \sim P$ and based on the data, we estimate some parametric (or non-parametric) functional of the underlying distribution $\theta^\star = \nu(P)$ by minimizing some loss function $\ell(\theta, X)$. A standard assumption for consistent estimation of $\theta^\star$ is that $\mathcal{R}(\theta) \triangleq \mathbb{E}[\ell(X, \theta)]$ is uniquely minimized at $\theta^\star$. In a time varying framework, we assume that the data generating distribution $P \equiv P_t$ changes with time and so does the parameter of interest $\theta_t^\star = \nu(P_t)$ along with the (time-varying) risk function $\mathcal{R}(\theta, t) \triangleq \mathbb{E}_{P_t}[\ell(X, \theta)]$.

As a concrete example, consider the object tracking problem studied in Patra et al. (2020): suppose we have installed $n$ sensors at positions $x_1, \ldots, x_n$ (which remain fixed over time) and let $\theta_t^\star$ be the location of the target object at time $t$. At each time, we get some noisy feedback from the sensors regarding the position of the target, i.e.

$$y_{i,t} = \|x_i - \theta_t^\star\|^2 + \epsilon_{it} \text{ for } 1 \le i \le n,$$

where $\epsilon_{it}$'s are iid (over $i$ and $t$) with mean zero and variance $\tau^2$. Denote by $\mathbf{y}_t \in \mathbb{R}^n$ to be the vector of observations from $n$ sensors at time $t$. A natural approach to estimate $\theta_t^\star$ is to minimize squared error loss $\ell(\mathbf{y}_t, \theta) = \sum_{i=1}^n (y_{i,t} - \|x_i - \theta\|^2)^2$, which yields the risk function

$$\mathcal{R}(\theta, t) = \sum_{i=1}^n \mathbb{E}[(y_{i,t} - \|x_i - \theta\|^2)] = n\tau^2 + \sum_{i=1}^n (\|x_i - \theta_t^\star\|^2 - \|x_i - \theta\|^2)^2.$$

From the risk function, it is immediate that under very mild assumptions on $x_1, \ldots, x_n$ (i.e. they are in general position, as discussed in Appendix A.1) we have

$$\theta^\star(t) = \arg \min_\theta \mathcal{R}(\theta, t), \ t \ge 0. \tag{2.1}$$

We will use the notations $\theta^\star(t)$ and $\theta_t^\star$ interchangeably to denote the same thing. From the above formulation, it is immediate that we are merely observing one sample/incident at each time point. Therefore if $\theta_t^\star$ behaves erratically over time, there is no hope to learn the evolution pattern from the data. Therefore, it is imperative to assume some smoothness on the target function $\theta_t^\star$. Our proposed method exploits the smoothness of $\theta_t^\star$ to improve its estimation over time.

We now formulate the problem: we assume that the underlying distribution function $\{P_t\}$ changes continuously with time $t$. As statisticians, we query the model at discrete time steps (i.e. say at time $\{kh\}_{k\in\mathbb{N}}$ where $h$ is the time step which controls the frequency of query) and observe $n$ samples $\{X_{it}\}_{1\le i\le n}$ from the distribution at that time. As a consequence, we have a sequential batch of data using which we aim to estimate $\theta_{kh}^\star$, i.e. the parameters at the time of query. Note that to estimate the parameter at time $t$ one may use all previous data points. We evaluate the quality of our estimator using the asymptotic tracking error (ATE) defined below.

**Definition 2.1** (Asymptotic tracking error (ATE)). *Let the true dynamic parameter $\{\theta_t^\star, \, t \geq 0\}$ be sequentially estimated as $\{\hat{\theta}_{kh}, k \in \mathbb{N}\}$ over the time grid $\{kh : k \in \mathbb{N}\}$, where $h > 0$ is the time step. Then the asymptotic tracking error is defined as*

$$ATE(\theta) = \limsup_{k \to \infty} \|\hat{\theta}_{kh} - \theta_{kh}^\star\|_2\,.$$

As mentioned in the Introduction, we here compare performance of a time-adjusted (PC) gradient descent method to a time-unadjusted (GD) one. As will be evident, both the methods require evaluation of certain derivatives of the risk function.

To motivate the predictor-corrector (PC) algorithm, we start by deriving the prediction correction term when the optimizer has access to the exact gradients of the (time-varying) cost function. The optimality of $\theta_t^\star$ implies

$$g(t) = \nabla_\theta \mathcal{R}(\theta_t^\star, t) = 0 \text{ for all } t. \tag{2.2}$$

Thus $g'(t) = 0$; *i.e.*

$$\nabla_{\theta\theta} \mathcal{R}(\theta_t^\star, t)\dot{\theta}_t^\star + \nabla_{\theta t} \mathcal{R}(\theta_t^\star, t) = 0, \tag{2.3}$$

where $\dot{\theta}_t^\star$ is the temporal drift of the optimal solution $\theta_t^\star$. We see that $\theta_t^\star$ satisfies the ODE:

$$\dot{\theta}_t^\star = -\nabla_{\theta\theta}^{-1} \mathcal{R}(\theta_t^\star, t) \nabla_{\theta t} \mathcal{R}(\theta_t^\star, t). \tag{2.4}$$

We interpret the right side of this ODE as a *prediction* of the change in $\theta_t^\star$. This suggests modifying the update rule of stochastic optimization algorithms to account for the predicted change in $\theta_t^\star$. This leads to the update rule

$$\hat{\theta}_{(k+1)h} = \hat{\theta}_{kh} - \eta \hat{\nabla}_\theta \mathcal{R}(\hat{\theta}_{kh}, kh) - h\{\hat{\nabla}_{\theta\theta} \mathcal{R}(\hat{\theta}_{kh}, kh)\}^{-1} \hat{\nabla}_{\theta t} \mathcal{R}(\hat{\theta}_{kh}, kh),$$

where $\eta > 0$ is a learning rate, $h$ is a (time) step, and $\hat{\nabla}_\theta \mathcal{R}(\theta, t)$, $\hat{\nabla}_{\theta\theta} \mathcal{R}(\theta, t)$, $\hat{\nabla}_{\theta t} \mathcal{R}(\theta, t)$ are estimates of $\nabla_\theta \mathcal{R}(\theta, t)$, $\nabla_{\theta\theta} \mathcal{R}(\theta, t)$, $\nabla_{\theta t} \mathcal{R}(\theta, t)$ respectively. We summarize the stochastic PC algorithm in Algorithm 1.

---

**Algorithm 1** Stochastic predictor-corrector based method

---

**Require:** step size $h > 0$, learning rate $\eta > 0$, estimated gradients $\hat{\nabla}_\theta \mathcal{R}(\theta, kh)$, Hessians $\hat{\nabla}_{\theta\theta} \mathcal{R}(\theta, kh)$ and time derivative of the gradient $\hat{\nabla}_{\theta t} \mathcal{R}(\theta, kh)$ for $k \in \mathbb{N}$

1: Initialize $\hat{\theta}_0$ at some value.
2: **for** $k \geq 0$ **do**
3:     Update $\hat{\theta}_{(k+1)h} = \hat{\theta}_{kh} - \eta \hat{\nabla}_\theta \mathcal{R}(\hat{\theta}_{kh}, kh) - h\{\hat{\nabla}_{\theta\theta} \mathcal{R}(\hat{\theta}_{kh}, kh)\}^{-1} \hat{\nabla}_{\theta t} \mathcal{R}(\hat{\theta}_{kh}, kh)$
4: **end for**

---

To motivate the benefit of accounting for the predicted change in $\theta_t^\star$, we present here a brief comparison of the tracking error of the PC algorithm with simple stochastic gradient descent without the correction term. For simplicity of exposition, we study the tracking error of these two algorithms in the non-stochastic setting. We begin by showing a lower bound on the tracking error of gradient descent. We suspect this result is known to experts, but it does not appear (to the best of our knowledge) in the literature.

**Theorem 2.2** (Lower bound). *There exists a $\mathcal{R}(\theta, t)$ that satisfies Assumption 3.1 such that the gradient descent algorithm with time step size $h > 0$ and learning rate $\eta > 0$ satisfies the following: there exists a $c > 0$ such that*

$$\liminf_{k \to \infty} \left\| \hat{\theta}_{kh} - \theta_{kh}^\star \right\|_2 \geq c\frac{h}{\eta}.$$

As we shall see in the subsequent section (see Theorem 3.3), the tracking error of the PC algorithm in the non-stochastic setting is $O(\frac{L''h^2}{M\eta})$. We restate this special case of Theorem 3.3 here for the reader's convenience.

**Corollary 2.3.** *(Non-stochastic version of Theorem 3.3) Assume that 3.1 and 3.2 holds and that $\mathcal{R}(\theta, t)$ is twice differentiable in both co-ordinates. Then with access to the true gradients, the sequence of estimates in Algorithm 1 with step size $h > 0$ and learning rate $\eta$ satisfies:*

$$\limsup_{k \to \infty} \left\| \hat{\theta}_{kh} - \theta_{kh}^* \right\| \leq \frac{L''h^2}{M\eta + hM'}\,,$$

*where $M' = \sup_{\theta, t} \|\{\nabla_{\theta\theta} \mathcal{R}(\theta, t)\}^{-1} \nabla_{\theta t} \mathcal{R}(\theta, t)\|_{op}$, $M = \sup_{t \geq 0} \|\nabla_\theta \mathcal{R}(\theta, t)\|$ and $L'' = \max_{t \geq 0} \frac{1}{2}\|\ddot{\theta}(t)\|$ are finite.*

To compare the rates for the PC algorithm and gradient descent, we set a learning rate such that $h/\eta \to 0$ as $h \to 0$. We see from Theorem 2.2 that the ATE for gradient descent ATE cannot converge faster than $h/\eta$. By comparison, the ATE for PC algorithm in the non-stochastic setting (see Theorem 2.3) converges to zero at rate $h^2/\eta$ (since it holds $\frac{L''h^2}{M\eta + hM'} \asymp \frac{L''h^2}{M\eta}$ as long as $h/\eta \to 0$), which a faster rate than $h/\eta$. Hence, we conclude that the ATE for predictor-corrector update converges at a faster rate than the ATE for gradient descent update. A high-level reasoning for such a distinction is the following: the gradient descent method does not consider or calibrate for the underlying smoothness in $\theta^\star(t)$ while performing the time updates, whereas predictor-corrector calibrates for the smoothness in the time update by adjusting the term $-h\{\nabla_{\theta\theta}\mathcal{R}(\hat{\theta}_t, t)\}^{-1}\nabla_{\theta t}\mathcal{R}(\hat{\theta}_t, t)$.

One of the main challenge in implementing the stochastic PC algorithm is obtain estimates of $\nabla_\theta\mathcal{R}(\theta, t), \nabla_{\theta\theta}\mathcal{R}(\theta, t), \nabla_{\theta t}\mathcal{R}(\theta, t)$:

1. **Estimation of gradient:** $\nabla_\theta\mathcal{R}(\theta, t)$ also appears in the SGD update rule; it is typically estimated with sample average approximation.
2. **Estimation of PC term:** $\nabla_{\theta\theta}\mathcal{R}(\theta, t), \nabla_{\theta t}\mathcal{R}(\theta, t)$ are quantities that arise due to the presence of the PC term in the stochastic PC update rule. Although it is possible to construct unbiased sample average approximations of them individually, obtaining an unbiased estimate of the overall PC term is generally not possible due to the presence of non-linearity: the PC term is a product of the inverse of the hessian matrix and the cross derivative with respect to the parameter and time. Fortunately, the stochastic PC algorithm is robust against biases in the estimate of the PC term. That said, naively estimating the PC term with sample average approximation can lead to non-convergence of the stochastic PC algorithm (see Remark 3.5 for details). In section 4, we present two ways of estimating/evaluating $\nabla_{\theta\theta}\mathcal{R}(\theta, t), \nabla_{\theta t}\mathcal{R}(\theta, t)$ that ensure the stochastic PC algorithm converges.

In the next section, we elucidate how errors in the estimates of $\nabla_\theta\mathcal{R}(\theta, t), \nabla_{\theta\theta}\mathcal{R}(\theta, t), \nabla_{\theta t}\mathcal{R}(\theta, t)$ affect the asymptotic tracking error of the stochastic PC algorithm.

## 3 THEORETICAL PROPERTIES OF THE STOCHASTIC PC ALGORITHM

We begin by stating our assumptions on the problem. We assume that the risk function is strongly convex with respect to $\theta$. This assumption implies that $\theta^\star(t)$ is uniquely identified at time $t$ and is crucial in studying the convergence of ATE.

**Assumption 3.1.** *$\mathcal{R}(\theta, t)$ is $\mu$-strongly convex with respect to $\theta$, i.e. for any $\theta_1, \theta_2$ and $t \geq 0$ it holds:*
$$\mathcal{R}(\theta_2, t) \geq \mathcal{R}(\theta_1, t) + (\theta_2 - \theta_1)^\top\nabla_\theta\mathcal{R}(\theta_1, t) + \frac{\mu}{2}\|\theta_2 - \theta_1\|_2^2.$$

We now assume that as a function of $t$, $\theta^\star(t)$ is smooth, which is naturally satisfied in numerous examples including dynamic least squares recovery, object tracking, *etc.*.

**Assumption 3.2.** *The function $\theta^\star \in \mathcal{C}_{\mathbb{R}^d}^2([0, \infty))$, i.e. $\theta^\star$ is twice continuously differentiable with respect to time and its double derivative is uniformly bounded over time.*

As will be seen in the subsequent theorems, the bounds on the ATE of these stochastic algorithms depend on the error in the estimation of the pertinent gradients, which are defined as follows:
$$\xi_t = \hat{\nabla}_\theta R(\theta, t) - \nabla_\theta R(\theta, t)$$
denotes the estimation error of the gradient and
$$\zeta_t = \hat{\nabla}_{\theta\theta} R(\hat{\theta}_{t_k}, t_k)^{-1}\hat{\nabla}_{\theta t} R(\hat{\theta}_{t_k}, t_k) - \nabla_{\theta\theta} R(\theta_{t_k}^*, t_k)^{-1}\nabla_{\theta t} R(\theta_{t_k}^*, t_k)$$
represents the error in the adjustment term for the temporal drift. We use $\sigma_\xi$ (resp. $\sigma_\zeta$) to denote an upper bound on $\sup_t \mathbb{E}[\|\xi_t\|]$ (resp. $\sup_t \mathbb{E}[\|\zeta_t\|]$). These bounds may or may not be a function of $h$, depending on the application. Note that we do not assume the estimates of the gradients are unbiased. Below we present our main theorems regarding the bounds on the ATE of stochastic gradient descent and 1 in terms of $\sigma_\xi, \sigma_\zeta$, learning rate $\eta$ and stepsize $h$:

**Theorem 3.3** (Stochastic predictor-corrector method). *The update sequence of stochastic predictor-corrector method presented in Algorithm 1 yields the follows bound on ATE:*
$$\limsup_{k \to \infty} \mathbb{E}[\|\hat{\theta}_{kh} - \theta_{kh}^\star\|] \leq \frac{L''h^2}{\eta M + hM'} + \eta\sigma_\xi + h\sigma_\zeta.$$

*for any small $\eta > 0$. When $\sigma_\xi$ and $\sigma_\zeta$ are independent of $h$ then choice of $\eta = h$ implies that ATE of stochastic predictor-corrector method is at-most of the order of $h$.*

Note that we don not require zero mean for the noise; we merely require them to have finite second moment. This is slightly more general that the usual stochastic optimization setting in which the noise is assumed to have mean zero. That said, in most of the applications that we have in mind, the noise comes from approximation of expectations by sample means, so the noise will be mean zero.

To see the benefits of the PC algorithm in the stochastic setting, we compare the ATE of the PC algorithm with that of stochastic approximation. Recall the stochastic gradient descent update rule:

$$\hat{\theta}_{(k+1)h} = \hat{\theta}_{kh} - \eta \hat{\nabla}_\theta \mathcal{R}(\hat{\theta}_{kh}, kh). \tag{3.1}$$

for some estimate of gradient. Its ATE is known (Cutler et al., 2021), but we restate it here to facilitate comparison:

**Theorem 3.4** (Stochastic gradient descent, Cutler et al. (2021))**.** *The update sequence of stochastic gradient method satisfies:*

$$\limsup_{k\to\infty} \mathbb{E}\big[\|\hat{\theta}_{t_k} - \theta_{t_k}^*\|\big] \leq \tfrac{Lh}{\mu\eta} + \eta\sigma_\xi$$

*for any small $\eta > 0$. Minimizing the right hand side with respect to $\eta$ yields:*

$$\limsup_{k\to\infty} \mathbb{E}\big[\|\hat{\theta}_{t_k} - \theta_{t_k}^*\|\big] \leq \sqrt{\tfrac{Lh\sigma_\xi}{\mu}} \, .$$

*Therefore, when $\sigma_\xi$ does not depend on $h$, the rate is $O(\sqrt{h})$.*

If we assume the error variances are independent of $h$, then simple stochastic gradient descent yields a bound of the order $\sqrt{h}$ on the ATE, whereas, the time-adjusted predictor-corrector based method yields a bound of the order of $h$, implying the superiority of the later for small stepsize. The superiority continues to hold even when the variances depend on $h$, as will be evident in the applications in the subsequent section.

**Remark 3.5** (Naive estimation of PC term fails)**.** *In practice, it is straightforward to obtain estimates of $\nabla_\theta \mathcal{R}(\theta, t)$ (e.g. sample average approximation), but it is less straightforward to estimate the cross-derivative term $\nabla_{\theta t} \mathcal{R}(\theta, t)$. A naive application of first-order finite differences to estimating the cross-derivative term leads to poor tracking performance because this estimate of the cross-derivative term leads to a $\sigma_\zeta$ term that is $O(h^{-1})$. Indeed, we have*

$$\hat{\nabla}_{\theta t} \mathcal{R}(\theta, kh) = \frac{\hat{\nabla}_\theta \mathcal{R}(\theta, kh) - \hat{\nabla}_\theta \mathcal{R}(\theta, (k-1)h)}{h}$$

$$= \frac{\nabla_\theta \mathcal{R}(\theta, kh) - \nabla_\theta \mathcal{R}(\theta, (k-1)h)}{h} + \frac{\xi_{kh} - \xi_{(k-1)h}}{h}.$$

*As long as $\mathcal{R}(\theta, t)$ is smooth with respect to $t$, the first term is $\nabla_{\theta t} \mathcal{R}(\theta, t) + O(h)$. But the second term is generally $O(h^{-1})$, e.g., if we assume that the errors over time are independent. Plugging this into the bound on the tracking error in Theorem 3.3, we see that the term that includes $\sigma_\zeta$ no longer depends on $h$, leading to a vacuous $O(1)$ bound. In Section 4, we use moving average schemes to obtain more accurate estimates of $\nabla_{\theta t} \mathcal{R}(\theta, t)$ to avoid this pitfall (e.g., for least square recovery problem in §4.1 see Equation (4.4) and Lemma 4.3 for it's estimation and error analysis).*

## 4 APPLICATIONS

In this section, we present three concrete time-varying optimization problems; all are characterized a gradual underlying distribution shift. This allows us to leverage the predictor-corrector (PC) method to improve tracking of the optimal trajectory. In some applications (e.g. the strategic classification example), there is a model for the distribution shift, so it is possible to evaluate the PC term exactly. In other applications, there is no such model, so it is necessary to approximate the PC term. We use a generic finite-difference approach in §4.1 and §4.3 to approximate the PC term. This approach is generally applicable, but it may not be optimal in applications in which the underlying distribution shift exhibits higher orders of smoothness.

### 4.1 LEAST SQUARES RECOVERY WITH FIXED DESIGN MATRIX

We first demonstrate the performance of the predictor-corrector method and compare it with gradient descent method in a linear regression model. We observe $\mathbf{y}_{kh} \triangleq \{Y_{kh,j}\}_{j=1}^{n}$ at time $kh$ for $k \in \mathbb{N}$ and some fixed stepsize $h$, where the observations are modeled as: $\mathbf{y}_t = \mathbf{X}\theta_t^\star + \boldsymbol{\epsilon}_t$. Here $\mathbf{X} \in \mathbb{R}^{n \times d}$ is a fixed time-invariant design matrix and the co-ordinates of $\boldsymbol{\epsilon}_t$ are i.i.d with mean $0$ and variance $\tau^2$. The parameter of interest here is the function $\theta^\star : [0, \infty) \to \mathbb{R}^d$. We consider a low dimensional scenario (i.e. $d < n$) and assume that the columns of $\mathbf{X}$ are in general position, i.e. $\mathbf{X}^\top \mathbf{X}$ is invertible. This immediately implies the following:

**Lemma 4.1.** *For any $t \in [0, \infty)$, $\theta_t^\star$ is the unique solution of the least square problem:*

$$\theta_t^\star = \arg\min_{\theta \in \mathbb{R}^d} \mathcal{R}(\theta, t) = \arg\min_{\theta \in \mathbb{R}^d} \frac{1}{2n} \mathbb{E}\big[\|\mathbf{y}_t - \mathbf{X}\theta\|^2\big]$$

*where $\mathcal{R}$ is the risk function and the expectation is taken with respect to the distribution of $\boldsymbol{\epsilon}$.*

The proof of Lemma can be found in Appendix A. We now compare the performance of stochastic gradient descent method (3.1) and PC method (Algorithm 1). The gradient of the risk function (with respect to) $\theta$ is: $\nabla_\theta \mathcal{R}(\theta, t) = \frac{1}{n}\mathbf{X}^\top (\mathbf{X}\theta - \mathbf{X}\theta_t^\star)$. We propose a *moving average based technique* to estimate the gradient of the risk function, *i.e.* for any time $t$ we define:

$$\hat{\nabla}_\theta \mathcal{R}(\theta, t) = \frac{1}{n}\mathbf{X}^\top \left(\mathbf{X}\theta - \sum_{i=0}^{m-1} \alpha_i \mathbf{y}_{t-ih}\right), \ \ \xi_t = \hat{\nabla}_\theta \mathcal{R}(\theta, t) - \nabla_\theta \mathcal{R}(\theta, t). \quad (4.1)$$

The optimal choice of the moving window length $m$ and $\{\alpha_i\}_{i=0}^{m-1}$ depends on a careful analysis of bias-variance trade-off of the estimation error $\xi_t$. First, note that $\xi_t$ can be decomposed into two terms as follows:

$$\xi_t = \frac{1}{n}\mathbf{X}^\top \mathbf{X} \left\{\theta_t^\star - \sum_{i=0}^{m-1} \alpha_i \theta_{t-ih}^\star\right\} - \frac{1}{n}\sum_{i=0}^{m-1} \alpha_i \mathbf{X}^\top \boldsymbol{\epsilon}_{t-ih} \triangleq A + B.$$

We can expand the bias term A via a two step Taylor expansion:

$$\theta_t^\star - \sum_{i=0}^{m-1} \alpha_i \theta_{t-ih}^\star = \theta_t^\star \left\{1 - \sum_{i=0}^{m-1} \alpha_i\right\} - h\dot{\theta}^\star(t)\sum_{i=0}^{m-1} i\alpha_i + \frac{h^2}{2}\sum_{i=0}^{m-1} i^2\alpha_i\ddot{\theta}^\star(\tilde{t}_i)$$

for some $\tilde{t}_i \in [t - ih, t]$. The following lemma presents an optimal scheme for choosing $\{m; \alpha_i, i = 0, \ldots, m - 1\}$:

**Lemma 4.2** (Gradient estimate). *For any fixed $m$, choosing the weights $\{\alpha_i\}$ as:*

$$(\alpha_0, \ldots, \alpha_{m-1}) \in \arg\min \left\{\sum_{i=0}^{m-1} a_i^2 : \sum_{i=0}^{m-1} a_i = 1, \text{ and } \sum_{i=0}^{m-1} ia_i = 0, a_i \in \mathbb{R}\right\} \quad (4.2)$$

*we obtain the following:*

1. *$\|A\|_2^2 = a_0^2 \times \mathcal{O}(m^4 h^4)$, where $a_0^2 = \max_{t \geq 0} \left\|\left(\frac{\mathbf{X}^\top \mathbf{X}}{n}\right) \ddot{\theta}^\star(t)\right\|_2^2$,*
2. *$\mathbb{E}[\|B\|_2^2] = b_0^2 \times \mathcal{O}(m^{-1})$, where $b_0^2 = \tau^2 \left\{\mathrm{TR}\left(\frac{\mathbf{X}^\top \mathbf{X}}{n^2}\right)\right\}$.*

*Therefore, taking $m = \mathcal{O}(h^{-4/5}\{a_0/b_0\}^{2/5})$ we have:*

$$\mathbb{E}[\|\xi_t\|_2] \leq \|A\|_2 + \{\mathbb{E}[\|B\|_2^2]\}^{1/2} = \mathcal{O}(a_0 m^2 h^2) + \mathcal{O}(b_0 m^{-1/2}) = \mathcal{O}(a_0^{1/5} b_0^{4/5} h^{2/5}). \quad (4.3)$$

The proof of this Lemma is deferred to the Appendix A. It is established in the proof that the optimal weights are: $\alpha_i = 2(2m - 1 - 3i)/\{m(m + 1)\}$. Using the error bound of Lemma 4.2 in Theorem 3.4 yields for small $\eta$, ATE is bounded by $\mathcal{O}(h/\eta) + \mathcal{O}(\eta h^{2/5})$. The right hand is minimized by taking $\eta = h^{3/10}$, which yields the order for asymptotic tracking error $\mathcal{O}(h^{7/10})$.

For **prediction-corrector based method** (Algorithm 1) we additionally need to estimate the Hessian and the time derivative of the gradient. The Hessian is constant $(\mathbf{X}^\top \mathbf{X}/n)$ and known (as we assume to know $\mathbf{X}$). To estimate $\nabla_{\theta t} \mathcal{R}(\theta, t) = -\{\mathbf{X}^\top \mathbf{X}/n\}\dot{\theta}_t^\star$, we again resort to a moving average based method, *i.e.* we set:

$$\hat{\nabla}_{\theta t} \mathcal{R}(\theta, t) = -\frac{1}{n}\mathbf{X}^\top \left\{\sum_{i=0}^{p-1} \beta_i \mathbf{y}_{t-ih}\right\} \quad (4.4)$$

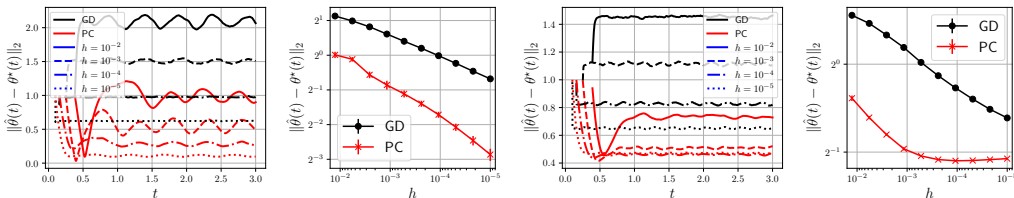

Figure 1: The two left-most figures show the performance of gradient descent and the PC method for time-varying linear regression. The left-most figure presents tracking error of GD and the PC methods for various choices of $h$ on the time interval $t \in [0, 3]$. The second figure from the left one compares the performance for a fixed $t = 3$ and different $h$ to illustrate how the performance of the methods vary with $h$. The two right-most figures show the performance of gradient descent and the PC method for object tracking. The second figure from the right shows the tracking error of the object tracking model for GD and the PC method for different choices of $h$. The $Y$-axis represents the tracking error and $X$-axis represents the time interval $t \in (0, 3)$.

for some choice of $p$ and the weights $\{\beta_j\}_{j=0}^{p-1}$, where the weights and the size of the window is obtained from a bias-variance trade-off:

$$\zeta_t = \left(\frac{\mathbf{X}^\top \mathbf{X}}{n}\right)^{-1}\left[\widehat{\nabla}_{\theta t}\mathcal{R}(\theta, t) - \nabla_{\theta t}\mathcal{R}(\theta, t)\right]$$
$$= -\sum_{i=0}^{p-1}\beta_i \theta_{t-ih}^\star + \dot{\theta}_t^\star - \left(\frac{\mathbf{X}^\top \mathbf{X}}{n}\right)^{-1}\frac{\mathbf{X}^\top}{n}\left(\sum_{i=0}^{p-1}\beta_i \boldsymbol{\epsilon}_{t-ih}\right) \triangleq C + D.$$

The following lemma presents the optimal choice of the weights, window length and consequently an error bound:

**Lemma 4.3.** *(Estimate of time derivative) For any $p$, if we choose the weights $\beta_i$'s as:*

$$(\beta_0, \ldots, \beta_{p-1}) \in \arg\min\left\{\sum_{i=0}^{p-1} b_i^2 : \sum_{i=0}^{p-1} b_i = 0, \text{ and } \sum_{i=0}^{p-1} ib_i = -\frac{1}{h}, b_i \in \mathbb{R}\right\} \quad (4.5)$$

*we have the following bounds on the error:*

1. $\|C\|_2^2 = \max_{t \geq 0} \|\ddot{\theta}^\star(t)\|_2^2 \times \mathcal{O}(h^2 p^2) \triangleq c_0^2 \times \mathcal{O}(h^2 p^2)$,
2. $\mathbb{E}[\|D\|_2^2] = \tau^2 \{\mathrm{TR}[(X^\top X/n)^{-1}]/n\} \times \mathcal{O}(h^{-2}p^{-3}) \triangleq d_0^2 \times \mathcal{O}(h^{-2}p^{-3})$.

*Setting $p = (d_0/c_0)^{1/4}h^{-3/4}$, we have:*

$$\mathbb{E}[\|\zeta_t\|_2] \leq \|C\|_2 + \{\mathbb{E}[\|D\|_2^2]\}^{1/2} = \mathcal{O}(c_0 hp) + \mathcal{O}(d_0 h^{-2}p^{-3}) = \mathcal{O}(c_0^{3/4}d_0^{1/4}h^{1/4}). \quad (4.6)$$

The $\beta_j$ in the above lemma takes the value $\frac{6(p-1-2j)}{p(p^2-1)h}$ as elaborated in the proof of the Lemma (see Appendix A). From the above lemma we have $\sigma_\zeta = \max_k \mathbb{E}[\|\zeta_{kh}\|_2] = \mathcal{O}(h^{1/4})$. Furthermore, we have established in the analysis of gradient descent method that the order of the error of the gradient estimation is $\sigma_\xi = \mathcal{O}(h^{2/5})$ (Lemma 4.2). Using these rates in Theorem 3.3 we conclude that predictor-corrector method with step size $\eta$ ATE is of the order $\mathcal{O}(h^2/\{\eta + h\}) + \mathcal{O}(\eta h^{2/5}) + \mathcal{O}(h^{5/4})$. The bound is minimized at $\eta = h^{4/5}$ and consequently, the order of the tracking error is $\mathcal{O}(h^{6/5})$. Therefore, predictor-corrector method yields faster rate (in terms of the step-size $h$) in comparison to the gradient based method.

**Simulation studies:** We compare the tracking performance of the gradient descent based and the predictor-corrector based method for the regression model via simulation. For simulation purpose, we set $d = 2, n = 40$. The values of $\{X_i\}_{i=1}^n$ are generated independently from $\mathcal{N}(0, I_2)$ and remain fixed over time. The true parameter is taken to be $\theta_t^* = (\sin(2\pi t), \cos(2\pi t))$ and at time $t$, the observation $\mathbf{y}_t \in \mathbb{R}^n$ is generated as $\mathbf{y}_t = \mathbf{X}\theta_t^* + \boldsymbol{\epsilon}_t$ where $\boldsymbol{\epsilon}_t \sim \mathcal{N}(0, 0.5I_n)$. Runtime analysis for the gradient descent based and the predictor-corrector based methods can be found in Figure 2 in Appendix C.

The tracking performance of the methods over a finite time interval ($t \in (0, 3)$) and the effect of $h$ on the limiting error (i.e. tracking error for some large $t$) are presented in Figure 1. In the left plot

of Figure 1, we observe that the predictor-corrector based method (denoted by PC) outperforms the gradient-descent based method (denoted by GD) in terms of the tracking error for all moderately large $t$ (here, $t \geq 1$) for all choices of $h \in \{10^{-2}, 10^{-3}, 10^{-4}, 10^{-5}\}$. As illustrated in our theory, the rate of convergence of limiting error of predictor-corrector based method decreases faster in $h$ compared to the gradient descent based method. The right side of Figure 1 establishes this phenomena through comparing the performances at $t = 3$ for several choices of $h$. As $\|\hat{\theta}_t - \theta_t^\star\|$ is a random quantity, error bars (over 10 Monte-Carlo iterations) are provided to quantify the variability of the tracking error, which turns out to be relatively small compared to the mean difference of the tracking error of the methods.

## 4.2 STRATEGIC CLASSIFICATION

In strategic classification, samples correspond to agents who change their features strategically to affect the output of the ML model (e.g. scammers modifying their scam to skirt a scam detector). To keep things simple, we assume the ML model is a binary classifier, and the positive output is advantageous for the agents. Following Hardt et al. (2016), we assume the agents are maximizing utility, and they have full information regarding the classifier. Thus an agent with features $x$ changes their features by solving the (expected) utility maximization problem

$$x_+(x) \triangleq \arg\max_{x' \in \mathcal{X}} u_+ f(x') - c(x, x'), \tag{4.7}$$

where $u_+ > 0$ is the utility from a positive output, $f(x)$ is the predicted probability of having positive output for $x$ from an ML model, $u_+ f(x)$ is the expected utility at $x$, and $c(x, x') > 0$ encodes the cost of changing features from $x$ to $x'$.

Let $p_1$ be the probability density function (pdf) of the positive class conditional at time $t$ and $f_t$ be the ML model deployed at time $t$. The agents respond strategically to $f_t$; *i.e.* an agent with feature $x$ changes their features to $x_+(x)$ (their label remains unchanged). The resulting change in the class conditional satisfies the continuity equation:

$$\partial_t p_1 + \nabla \cdot (v p_1) = 0,$$

where $\nabla \triangleq [\partial_{x_1} \quad \ldots \quad \partial_{x_d}]$ is the spatial gradient operator and $v$ is the vector field $v(x) \triangleq x_+(x) - x$. Similarly, the pdf of the negative class conditional also satisfies the continuity equation.

This change in the distribution of agent features leads to a time-varying optimization problem:

$$\theta(t) \in \arg\min_\theta f(\theta, t) \triangleq \pi \int_{\mathcal{X}} \ell(f_\theta(x), 1) dP_1(x) + (1 - \pi) \int_{\mathcal{X}} \ell(f_\theta(x), 0) dP_0(x),$$

where $\pi \triangleq \mathbb{P}\{Y = 1\}$ is the faction of the positive class and $\ell$ is a loss function for the classification task. Interchanging limits freely, we see that it is possible to estimate $\nabla_\theta f(\theta, t)$ and $\nabla_{\theta\theta} f(\theta, t)$ empirically:

$$\nabla_\theta f(\theta, t) = \pi \int_{\mathcal{X}} \nabla_\theta \ell(f_\theta(x), 1) dP_1(x) + (1 - \pi) \int_{\mathcal{X}} \nabla_\theta \ell(f_\theta(x), 0) dP_0(x),$$
$$\nabla_{\theta\theta} f(\theta, t) = \pi \int_{\mathcal{X}} \nabla_\theta^2 \ell(f_\theta(x), 1) dP_1(x) + (1 - \pi) \int_{\mathcal{X}} \nabla_\theta^2 \ell(f_\theta(x), 0) dP_0(x)$$

Similarly, it is possible to estimate $\nabla_{\theta t} f(\theta, t)$ empirically:

$$\nabla_{\theta t} f(\theta, t) = \pi \int_{\mathcal{X}} \ell(f_\theta(x), 1) \partial_t p_1(x, t) dx + (1 - \pi) \int_{\mathcal{X}} \ell(f_\theta(x), 0) \partial_t p_0(x, t) dx$$
$$= -\pi \int_{\mathcal{X}} \ell(f_\theta(x), 1) \nabla \cdot (v(x) p_1(x, t)) dx - (1 - \pi) \int_{\mathcal{X}} \ell(f_\theta(x), 0) \nabla \cdot (v(x) p_0(x, t)) dx$$
$$= \pi \int_{\mathcal{X}} \nabla_x \ell(f_\theta(x), 1)^\top v(x) p_1(x, t) dx + (1 - \pi) \int_{\mathcal{X}} \nabla_x \ell(f_\theta(x), 0)^\top v(x) p_0(x, t) dx.$$

where we appealed to the continuity equation in the second step and Green's identities in the third step. Unlike the other two applications in this section, it is possible to compute the PC term exactly (without resorting to finite-difference approximation) here.

## 4.3 OBJECT TRACKING

Our third application is object tracking problem proposed and analyzed by Patra et al. (2020). Assume we have $n$ sensors placed the position $\{X_i\}_{i=1}^n$ in $\mathbb{R}^d$ and $\theta^\star(t)$ denotes position of the object (that we aim to track) at $t$. At any given point, we observe a noisy version of some monotone function of distance of the object from the sensors, i.e. we observe:

$$Y_{i,t} = f(\|X_i - \theta_t^\star\|^2) + \epsilon_{i,t} \ \forall \ 1 \leq i \leq n .,$$

In this example, we assume we know $f$. The case of unknown $f$ is more complicated and beyond the scope of the paper. The risk function for estimating $\theta_t^\star$ under quadratic loss function is:

$$\mathcal{R}(\theta, t) = \mathbb{E}\Big[\frac{1}{2n}\sum_{i=1}^n \big(Y_{i,t} - f(\|X_i - \theta\|^2)\big)^2\Big] = \frac{\sigma^2}{2} + \frac{1}{2n}\sum_{i=1}^n \big(f(\|X_i - \theta\|^2) - f(\|X_i - \theta_t^\star\|^2)\big)^2$$

Note that the risk function here is not strongly convex (so it does not satisfy the assumptions of Theorem 3.3), but as we shall see, the PC algorithm nevertheless outperforms standard first-order methods. The gradient and the Hessian, that are required for GD and PC methods, can be easily estimated as using sample averages (exact expressions are presented in the Appendix B). The time derivative of the gradient is:

$$\nabla_{\theta,t}\mathcal{R}(\theta, t) = -\frac{2}{n}\sum_{i=1}^n f'(\|X_i - \theta\|^2)(\theta - X_i)\frac{d}{dt}f(\|X_i - \theta_t^\star\|^2)$$

To estimate $\frac{d}{dt}f(\|X_i - \theta_t^\star\|^2)$ we again resort to the moving average procedure:

$$\widehat{\partial}_t f(\|X_i - \theta_t^\star\|^2) = \sum_{j=0}^{p-1}\beta_j Y_{i,t-jh}\,,$$

where $p$, $\{\beta_j\}_{j=1}^p$ are chosen carefully to balance the bias variance trade-off. For notational simplicity we drop the index $i$ and define $g(t) = f(\|X_i - \theta_t^\star\|^2)$. From the relation $Y_{i,t} = g(t) + \epsilon_{i,t}$ and Assumption 3.2, we have:

$$\sum_{j=0}^{p-1}\beta_j Y_{i,t-jh} = \sum_{j=0}^{p-1}\beta_j\{g(t) - jhg'(t) + \frac{j^2h^2}{2}g''(\tilde{t}_j)\} + \sum_{j=0}^{p-1}\beta_j\epsilon_{i,t-jh}$$

Now if the sequence $\{\beta_j\}_0^{p-1}$ satisfies $\sum_j \beta_j = 0$ and $\sum_j j\beta_j = -(1/h)$ then we have:

$$\sum_{j=0}^{p-1}\beta_j Y_{i,t-jh} - g'(t) = (\sum_j j^2\beta_j)O(h^2) + \sum_{j=0}^{p-1}\beta_j\epsilon_{i,t-jh}\,.$$

The variance of the error term is $\sigma_\epsilon^2\sum_j\beta_j^2$. Therefore we will choose $\{\beta_j\}$ and $p$ by minimizing $\sum_j\beta_j^2$ subject to the above constraints. Similar calculation as of Example 1 (Lemma 4.3) yields $p = \mathcal{O}(h^{-3/4})$ and $\{\beta_j\}_{j=0}^{p-1}$ is $\beta_j = \frac{6(p-1-2j)}{p(p^2-1)h}$.

**Simulation study**   We consider $n = 11^2$ sensors placed at $\{-1, -0.8, \ldots, 0.8, 1\}^2 \subset [-1, 1]^2$ grid. The moving object takes the path $\theta_t^\star = (\sin(2\pi t), \cos(2\pi t))$ and we let $f(x) = x$. We generate noisy observations of the object as $y_{it} = \|X_i - \theta_t^\star\|^2 + \epsilon_{it}$ where $\epsilon_{it} \sim \mathcal{N}(0, 1/4)$. Runtime analysis for the two methods can be found in Figure 2 in Appendix C. Figure 1 (third and forth plots from left) presents the tracking error of both the methods (gradient descent based method (resp. predictor-corrector based method) is represented as GD (resp. PC)) for four choices of $h \in \{10^{-2}, 10^{-3}, 10^{-4}, 10^{-5}\}$. The superiority of the performance of the PC method for all $t \geq 1$ and for various choices of $h$ is evident from the third picture from the left-side of Figure 1. In the last picture, we compare the limiting performance of both the methods (here we take the error at $t = 3$ to be the limiting error) for several choices of $h$. The superiority of the performance of PC corroborates our theoretical finding that the tracking error for PC converges faster in terms of $h$ compared to the GD. We also show error bars (over 10 Monte-Carlo iterations) of the random tracking error $\|\hat{\theta}_t - \theta_t^\star\|$ at $t = 3$ on the right picture, which indicates that the variability is relatively small compared to the difference of the mean tracking error of the methods.

## 5   CONCLUSION

We developed predictor corrector algorithms for stochastic time-varying optimization. These algorithms leverage smoothness in the temporal drift to anticipate changes to the optimal solution. We showed that these algorithms have smaller asymptotic tracking errors than their non-predictor corrector counterparts and demonstrated their efficacy in three applications. Although we focused on first-order algorithms in this paper, the predictor corrector term in the update rules of our PC algorithms can be incorporated into the update rules of other algorithms (*e.g.* Newton-type methods). We hope that the benefits of first-order PC algorithms motivates others to study PC versions of other algorithms.

ACKNOWLEDGMENTS

This paper is based upon work supported by the National Science Foundation (NSF) under grants no. 2027737 and 2113373.

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
