# OpenReview forum: "Predictor-corrector algorithms for stochastic optimization under gradual distribution shift"
_ICLR.cc/2023/Conference — ICLR 2023 poster_

### Official Review · Reviewer_SvER · 2022-10-24

**Confidence:** 4
**Clarity, Quality, Novelty And Reproducibility:** No concerns
**Correctness:** 4
**Technical Novelty And Significance:** 4
**Empirical Novelty And Significance:** 4
**Recommendation:** 6

**Strength And Weaknesses:**

The proposed approach makes a lot of sense and is presented clearly and intuitively. I agree with the authors that most approaches to tracking tend to ignore any possible temporal structure in the losses, and exploiting this seems a promising direction.

The requirement for strong convexity seems a bit limiting, however. This seems mostly needed for tracking the parameter vector distance, but one might expect that in many cases we are not interested in $$\|\theta -\theta^\star\|$$ but $$R(\theta)-R(\theta^\star)$$. Do the described techniques have any relevance for this metric?

Moreover, modern analysis of gradient descent methods typically focuses on non-asymptotic convergence bounds. Is there any way to obtain such a bound here?

The paper could have been augmented with a discussion of lower bounds - is there any way in which the described techniques are optimal?

Finally, the authors may wish to compare with the literature on dynamic regret in online convex optimization, which may provide another alternative to gradient descent which has seen a lot of recent development. Since these techniques also ignore temporal structure in the losses, I would expect this method to still compare favorably.


**Summary Of The Paper:**

This paper considers the problem of tracking a drifting optimization target under the assumption that the loss function is a smooth function of both the optimization parameter and time. The key insight is to compute the derivative of the minimizer with respect to time by differentiating the loss function with respect to time. Gradient descent is then augmented with an estimate of this derivative in order to more closely track the target.

High level calculations show that with sufficiently good estimates, this method will track the target with a better dependence on the time interval between updates ($h\rightarrow h^2$ where $h$ is the time interval).

In any given example, one then needs to build good estimates of the derivative of the target with respect to time. The authors explore several specific examples in which this can be obtained.


**Summary Of The Review:**

The paper initiates a study of drifting optimization utilizing temporal structure in the loss function. Results are presented that can improve over naive gradient descent, along with experimental validation on certain problems.

The theoretical development seems promising, although perhaps a little more limited than ideal, establishing some cases in which the techniques can improve over SGD in certain cases, but without many general convergence bounds.

---

> ### Author Response · Authors · 2022-11-17
> **Response to reviewer SvER**
>
> Thank you for the comments. We address your concerns below.
>
> **Relaxing strong convexity:** In statistical estimation problem, typically we don't need strong convexity if we only care about the prediction consistency, i.e. $R(\theta) - R(\theta^\star)$. However, in case of time-varying stochastic optimization problems, we need to update $\theta$ explicitly even if we want to bound $R(\theta) - R(\theta^\star)$. In our stochastic predictor-corrector algorithm (Algorithm 1), we need to calculate the inverse of the double derivative of the risk function to adjust for the time drift (step 3). This term is crucial as it makes our algorithm more efficient than the standard gradient descent based methods by taking care of the temporal changes. Therefore we require strong convexity for our method for this second order correction. It may be possible to use a generalized inverse if the hessian of the risk function is not invertible, which is an interesting avenue for future research.
>
> **Non-asymptotic bound:** Following your suggestion, we have established a non-asymptotic result on the linear convergence of the noiseless pc algorithm in Appendix D (Theorem D.1). The expression in the theorem contains an additional error term of the order $h^2$, which appears as the true parameter $\theta^\star(t)$ is itself dynamic and consequently we cannot avoid the discretization error.
>
> **Lower bound:** Following your suggestion, we have established a matching lower bound for the PC algorithm in the noiseless setting. Kindly see Theorem A.3 in Appendix A.2 in the revised version of the manuscript. Consequently, we now have lower bounds for both the methods, one with the correction and one without the correction term.
>
> **Application to dynamic regret:**  At the very core, *our tracking problem has a close connection to a dynamic regret problem*. In the dynamic regret framework one aims to minimize the dynamic regret: $T^{-1}\sum_{j=1}^T\\{f_t(\theta_t) - f_t(\theta^\star_t)\\}$, where $\theta^\star_t$ is the minimizer of $f_t$. In the time varying optimization, the goal is closely related: tracking the dynamic parameter $\theta^\star(t)$, where $f_t$ is $R(\theta, t)$. Therefore, as you pointed out, our theory suggest that the PC algorithm should outperform standard gradient based method (which does not account for the temporal drift) in other dynamic regret problems. However, this is currently out of the scope of this paper and definitely an interesting avenue for further research.

---

### Official Review · Reviewer_iKWJ · 2022-10-25

**Confidence:** 2
**Correctness:** 3
**Technical Novelty And Significance:** 2
**Empirical Novelty And Significance:** Not applicable
**Recommendation:** 5

**Clarity, Quality, Novelty And Reproducibility:**

The paper is well written in general, and the results seem novel. But I raised the concerns above and some explanation on the matter is desirable.

**Strength And Weaknesses:**

Strength
- simple update rule with a non-trivial theoretical guarantee
- promising experimental results


Weakness
- some technical issues are not clear (see below)

As summarized above, the claimed technical contributions are non-trivial and solid. However, I have several concerns on technical details, which could affect the evaluations of the theoretical results.

* The sample complexity or success probability are not discussed.
The problem setting is stochastic and the goal is to minimize ATE, i.e., the distance between the proposed and true parameters. To minimize this, the statistician has only access to finite samples from the distributions, which causes estimation errors. So, I am strongly wondering how the authors obtain the claimed results while avoiding these sampling/estimation issues. The paper needs some explanation on this matter to justify the results. Or some implicit assumptions exist.

So far, I do not have an intuition that the claimed results are correct.

**Summary Of The Paper:**

The paper considers the problem of tracking the sequence of best parameters of quadratic functions, where the best parameters changes in a stochastic way. The paper proposes an algorithm called PC and proves its convergence guarantees in terms of ATE. Then, some application problems are introduced, and the PC algorithm is applied. Experimental results show the advantages of PC against the SGD.

**Summary Of The Review:**

The theoretical and empirical results seem non-trivial, but I have some concerns about details of theoretical results.

---

> ### Author Response · Authors · 2022-11-17
> **Response to reviewer iKWJ**
>
> We thank the reviewer for the comments. We address your questions and concerns below.
>
> **Sample complexity:** You have rightly pointed out that the setting is stochastic and that is precisely why our results are for **expected** tracking error instead of the true tracking error. This expectation is taken over the noise distribution. The key reason for bounding the expected error is as follows: in online learning problem or time varying optimization, it is typically assumed that we only have access to a noisy version of derivatives of the true function. The source of noise can be anything and often is not taken into account directly. A special case is statistical learning problem, where the noise is introduced through the empirical version of the risk function. Therefore, in that particular context, it makes sense to present the result in terms of number of samples, as that quantifies the noise level (more samples mean less variance of the loss function). However if the noise source is something different, then the noise level may not be expressed via the number of samples. To avoid this issue, researchers tend to express these type of results in terms of expectation, so that the bound depends only on the noise variance and consequently covers a wide array of scenarios.
>
> To understand the difference between noisy case and the noiseless case, consider Corollary 2.3 and Theorem 3.3. In Theorem 3.3 we have derived the limit of the **expected value of the tracking error** (as we have noise), whilst in Corrolary 2.3 (which is the noiseless setting), our limit **does not involve any expectation**.
>
> In most examples of dynamic systems, it is expected that the noise level is fixed and does not increase to infinity. One rather assumes that the queries are made at finer time windows, i.e., the length of the time intervals (denoted by $h$) decreases to zero and studies the tracking error as a function of $h$. We also follow this paradigm; we assume a fixed number of samples/noise variance at each time point and study the ATE for GD and PC updates in terms of $h$.
>
> **Theoretical intuition:** To understand the implications of our theory, let us consider the main theorem of the paper, Theorem 3.3, which provides an upper bound on the expected asymptotic tracking error of the stochastic predictor corrector method. The bound has three terms; the second and the third term quantifies the effect of the noise in the gradient and the correction term respectively (recall that in our update rule, we have both the gradient term and a correction term for temporal drift, see Step 3 of Algorithm 1). If there is no noise, then the second and the third term will be zero, which corresponds to the noiseless setting (Corollary 2.3).
>
> The first term therefore effectively quantifies the error in the noiseless setting in terms of h and $\eta$. To understand the implication, let us assume that we observe the true derivatives at each time points. Fixing $\eta$, if $h$ goes to $0$, then our error goes to $0$. This is intuitively true as small $h$ means we are effectively observing the entire continuous process and consequently the tracking error is negligible. On the other hand, if $\eta$ goes to $0$ for a fixed $h$, then error becomes of the order $h$. This is because, when $\eta$ is very small, we are only performing the time correction step but are *not* minimizing the risk by moving along the opposite direction of the gradient. Therefore, our prediction is always proportional to the time step owing to the continuity of the process.
>
> In a nutshell, our theory provides a bound on the expected tracking error in terms of the time gap (h), learning rate ($\eta$) and noise variances ($\sigma_\zeta, \sigma_\xi$). The error is proportional to the noise variance and time gap and inversely proportional to the learning rate in absence of noise. In the revised manuscript, we have also provided a matching lower bound (Theorem A.3 in Appendix A.2) for the noiseless case when $\eta > h$.

---

### Official Review · Reviewer_YXuw · 2022-10-26

**Confidence:** 2
**Correctness:** 3
**Technical Novelty And Significance:** 3
**Empirical Novelty And Significance:** 3
**Recommendation:** 5

**Clarity, Quality, Novelty And Reproducibility:**


"The rest of the paper is organized as follows" -- and then only sections 2 and 4 are described. What about sections 3 and 5?

let \theta is -> let \theta be

"i.e. they are in general position" -> ? What does it mean for things to be in "general position"?

"it is immediate" -> ? (I don't know what is meant here.)

interchangeably to denote the same thing -> interchangeably

" on target function" -> on the target function

"it’s estimation" -> its estimation

"As statisticians, we query/intervene the model at discrete time steps" -> ? (this sounds like you are saying all statisticians do this)

"As mentioned in the Introduction, we here compare performance of a time-adjusted gradient descent method to a time-unadjusted one"  -- The introduction did not mention this. The introduction mentioned comparing a predictor-corrector based algorithm versus gradient descent.

"demonstrated their efficacy in three applications" -- I am not sure about this. These are toy experiments in simulated data, not "applications" by my definition of the word.


**Strength And Weaknesses:**

The paper is somewhat interesting, and appears to be quite thorough, but relatively unconvincing to me, since it is mostly theoretical, and the only experiments are in toy synthetic tasks/settings. For me this is particularly surprising because so much of the text talks about very practical issues, like distribution shift, and object tracking. In fact the paper is not doing any normal kind of tracking, and instead considers a simple synthetic problem where "sensors" return estimates of the target's "distance", with some zero-mean variance. Maybe this paper will be convincing and helpful to some, but in my case I don't know what to do with it.  Since the paper is out of my domain I certainly won't fight to reject it, but I hope my suggestions on language will be somewhat helpful.

**Summary Of The Paper:**

This paper presents a method to incorporate some knowledge about gradual distribution shifts during online gradient descent. The key idea is to add a predictor-corrector term to the gradient (i.e., adding an approximation of the time derivative), which tries to take into account the fact that the expected error changes over time with the current parameters.


**Summary Of The Review:**

This paper may be interesting to some, but given the language on handling distribution shift and offering improvements to object tracking, I expected something more practical, rather than assumptions/lemmas/theorems and toy experiments in simulated data.

---

> ### Author Response · Authors · 2022-11-17
> **Response to reviewer YXuw**
>
> We thank the reviewer for the detailed feedback and comments. We address questions and concerns below.
>
> **Practical examples:** We agree with you that application of our predictor corrector method to some practical examples can add more insights in terms of applicability, and it is one of our on-going researches. However, the main aim of this paper is to provide theoretical insights: particularly we show that the correction term can improve upon the performance of the online gradient descent algorithm for several ubiquitous statistical models.
>
> **Clarity, Quality, Novelty And Reproducibility:** Thank you for pointing out the typos and we have corrected them in our revision. Please see our responses to the other comments.
>
> - **What about sections 3 and 5:** We have added the references to Sections 3 and 5.
>
> - **General position:** It means any $k \in {2, \dots, d+1}$ collection of points from $x_1, \dots, x_n\in \mathbb{R}^d$ is affine independent, whose definition follows:  $y_1, \dots, y_k$ are affine independent if only solution of $\alpha_1, \dots, \alpha_k\in \mathbb{R}$ to the equations (1) $\sum_{i = 1}^k \alpha_i y_i = 0$ and (2) $\sum_{i = 1}^k \alpha_i = 0$ is $\alpha_1 = \dots = \alpha_k = 0$. We have now added a formal definition of general position in Appendix A.1.
>
> - **It is immidiate:** We have added a discussion in Appendix A.1 related to this and the previous comment.
>
> - **As statisticians, we query/intervene the model at discrete time steps:**  In a study of dynamic system it is practically impossible to query data smoothly over time and the only choice is to query data over some discrete times.
>
> - **As mentioned in the Introduction, we here compare performance of a time-adjusted gradient descent method to a time-unadjusted one.** By *time-adjusted gradient descent* method, we refer to the gradient update with predictor-corrector term and by *time-unadjusted one* we refer to regular online gradient descent method. The reason behind this naming is that in predictor-corrector method, we add an additional term (the term involving hessian and time derivative) to the standard gradient update rule to take care of the temporal drift. We have clarifier this in our revision.

---

### Official Review · Reviewer_UaJ8 · 2022-10-27

**Confidence:** 2
**Correctness:** 3
**Technical Novelty And Significance:** 3
**Empirical Novelty And Significance:** 3
**Recommendation:** 6

**Clarity, Quality, Novelty And Reproducibility:**

This is in general a high quality paper, with clear presentations, and some novel ideas. There are some important questions which this work does not address, though.

**Strength And Weaknesses:**

Strengths:
1. This work focuses on an important and interesting question in ML community.
2. The proposed solution (adding an additional predictor-correct term to the gradient updating rule) is intuitive, general, and easy to implement.
3. Some theoretical insights are given to demonstrate the usefulness of the proposed algorithm, together with some concrete applications.

Weakness:
It seems to me this work does not provide a general way to estimate the predictor-correct term. Instead, only some specific forms for the applications in Section 4 are discussed. For the proposed algorithm to be truly useful, I wonder how easy it is for a practitioner to come-up with an estimation for the term, or what would be the general recommendation for them. Or maybe, another question whether the proposed moving average algorithm can be shown to be useful in for general applications.

**Summary Of The Paper:**

This paper presents a novel predictor-corrector algorithm for time-varying stochastic optimization. It exploits the continuity of the domain shift. This work demonstrates the superiority of the proposed algorithm over stochastic gradient descent both theoretically and empirically. Further, this works shows that a simple sample average estimation of the predictor-corrector term may lead to non-convergence. In applications shown, a moving average based method is instead applied to estimate the predictor-corrector term. The utility of the proposed method is shown in three concrete real applications.

**Summary Of The Review:**

I think this paper targets a very important question and provides direction for an intuitive solution. Yet, there are some important parts missing from its current form which somehow prevents it from being more useful and practical.

---

> ### Author Response · Authors · 2022-11-17
> **Response to reviewer UaJ8**
>
> We thank the reviewer for the comments. We address your concern below.
>
> **Moving average algorithm in general applications:** We expect the moving average schemes to generalize to applications with fixed design matrix, whose empirical investigations are interesting questions for future study. For a more general setup (beyond fixed design) one may use our weighting scheme with the moving average windows set at $m = O(h^{-\gamma_m})$ and $p = O(h^{-\gamma_p})$, where $\gamma_m, \gamma_p>0$ are problem specific hyperparameters. These hyperparameters can be chosen via cross-validation.

---

### Author Response · Authors · 2022-11-17
**General response**

We thank all the reviewers for carefully reading our paper and for their insightful comments. Below we summarize the main changes we have incorporated in our revised draft:

- In Theorem A.3 of Appendix A.2 we have established a lower bound for the tracking error of noiseless predictor-corrector algorithm. The lower bound matches with the upper bound when $\eta \gtrsim h$, which is the more interesting regime, in the other regime (i.e. $\eta \lesssim h$) predictor-corrector method does not outperform gradient descent based method.

- We have also provided a non-asymptotic linear convergence for the noiseless predictor-corrector algorithm (Theorem D.1 in Appendix D) upto and error of order $h^2$. This error is unavoidable which appears through the discretization of the countinuous process.

---

### Decision · Program_Chairs · 2023-01-20

**Decision:**

Accept: poster

**Justification For Why Not Higher Score:**

see meta-review above - some weaknesses in this submission.

**Justification For Why Not Lower Score:**

see meta-review above - the ideas were deemed interesting and valuable for the community to be exposed to.

**Metareview: Summary, Strengths And Weaknesses:**

This was a borderline paper, and there was a constructive and lively discussion among the reviewers.

Some of the weaknesses of this paper were the absence of non-asymptotic results (all are in the limit of infinitely many steps), and difficulties in estimating the predictor-correct term (even in simple experiments).

However, the reviewer lauded the creativity and novelty of the ideas exposed in the paper - as well as the quality of exposition and motivation.

**Note From Pc:**

if the above contains the word "oral" or "spotlight" please see: "oral" presentation means -> notable-top-5% and "spotlight" means -> notable-top-25%. As stated in our emails, we are disassociating presentation type from AC recommendations

**Summary Of Ac-Reviewer Meeting:**

see meta-review above - it reflects the discussion during the AC-reviewer meeting.